# Efficient Generative Model Training via Embedded Representation Warmup

## Abstract

Generative models face a fundamental challenge: they must simultaneously learn high-level semantic concepts (what to generate) and low-level synthesis details (how to generate it). Conventional end-to-end training entangles these distinct, and often conflicting objectives, leading to a complex and inefficient optimization process. We argue that explicitly decoupling these tasks is key to unlocking more effective and efficient generative modeling. To this end, we propose **E**mbedded **R**epresentation **W**armup **(ERW)**, a principled two-phase training framework. The first phase is dedicated to building a robust semantic foundation by aligning the early layers of a diffusion model with a powerful pretrained encoder. This provides a strong representational prior, allowing the second phase—generative full training with alignment loss to refine the representation—to focus its resources on high-fidelity synthesis. Our analysis confirms that this efficacy stems from functionally specializing the model's early layers for representation. Empirically, our framework achieves a **11.5×** speedup in **350 epochs** to reach **FID=1.41** compared to single-phase methods like REPA (Yu et al., 2024).

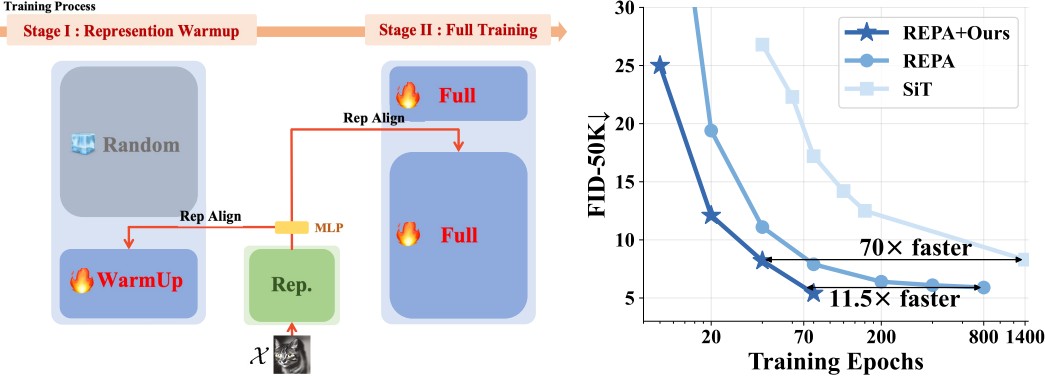

Figure 1: **A Staged Approach: First Build Semantics, Then Synthesize.** Our framework operationalizes the decoupling of semantic understanding from generative synthesis. In **Phase 1 (Semantic Foundation)**, we exclusively train the model's early layers to align with a pretrained encoder (e.g., DINOv2 (Oquab et al., 2023)), establishing a robust understanding of *what* to generate. In **Phase 2 (Guided Synthesis)**, the full model is trained. The plot empirically demonstrates the power of this decoupling: ERW converges dramatically faster and achieves superior performance compared to single-phase training like REPA (Yu et al., 2024), which entangles both learning tasks.

## 1 Introduction

> *"All roads lead to Rome, but it is not as good as being born in Rome."*

Deep generative models, particularly diffusion models (Ho et al., 2020; Song et al., 2020), have achieved remarkable success in high-fidelity image generation. These models excel at tasks ranging from unconditional image generation (Dhariwal & Nichol, 2021) to text-to-image synthesis (Ramesh et al., 2022; Saharia et al., 2022), demonstrating a profound capacity to model complex data distributions. However, underpinning their impressive capabilities is a fundamental tension, arising from a multitude of *entangled learning objectives*.

At its core, effective generation requires both *semantic understanding*—comprehending what constitutes meaningful content—and *visual synthesis*—translating abstract concepts into precise pixel-level details. Conventional end-to-end training entangles these objectives within a single optimization process, forcing the model to concurrently learn high-level conceptual knowledge and low-level rendering skills. This entanglement creates inherent optimization conflicts, a challenge reminiscent of the classic perception-distortion trade-off (Blau & Michaeli, 2018). Early in training, the model's attempts to fit pixel-level details may interfere with its ability to capture global semantic structures, an issue exacerbated by the known spectral bias of neural networks towards learning low-frequency components first (Rahaman et al., 2019; Sauer et al., 2021). Consequently, later stages may struggle to refine generation quality due to inadequate representational foundations.

Recent studies have begun to acknowledge this tension. While diffusion models implicitly learn semantic features during denoising (Yang & Wang, 2023; Xiang et al., 2023), these representations often lack the robustness and versatility of dedicated self-supervised approaches (Caron et al., 2021; Oquab et al., 2023). Moreover, Kadkhodaie & Simon (2024) highlight the critical bottleneck between memorizing semantic information and generalizing to realistic distributions. Methods like REPA (Yu et al., 2024) have attempted to address this by aligning diffusion representations with pretrained encoders throughout training, yet they still suffer from the fundamental challenge of joint optimization. These observations lead us to a pivotal question:

> ℚ: **Can we fundamentally simplify generative model training by decoupling semantic understanding from visual synthesis, thereby allowing each component to be optimized more effectively?**

Self-supervised learning approaches, including contrastive methods (Chen et al., 2020a), masked autoencoders (He et al., 2022), and recent advances like DINOv2 (Oquab et al., 2023), have demonstrated exceptional capabilities in learning rich semantic representations. However, effectively integrating these external representations into diffusion models remains challenging due to fundamental mismatches: diffusion models operate on progressively noisy inputs while self-supervised encoders are trained on clean data, and architectural differences further complicate direct integration.

**Our approach.** We propose that the key to resolving this challenge lies in explicitly *decoupling* the learning of semantic understanding from visual synthesis. To this end, we introduce Embedded Representation Warmup **(ERW)**, a principled two-phase framework that operationalizes this decoupling philosophy. Our approach is grounded in the observation that diffusion models naturally exhibit a functional specialization: early layers predominantly handle semantic processing (what we term the *Latent-to-Representation* or L2R circuit), while later layers focus on generative refinement (the *Representation-to-Generation* or R2G circuit).

Rather than forcing both circuits to learn simultaneously from scratch, ERW strategically separates their optimization: **Phase 1 (Semantic Foundation)** establishes a robust semantic foundation by dedicating training exclusively to aligning the L2R circuit with a pretrained self-supervised encoder (e.g., DINOv2). This phase ensures the model is "born in Rome"—equipped with mature semantic understanding from the outset. **Phase 2 (Guided Synthesis)** then leverages this foundation to focus training resources on the R2G circuit, optimizing visual synthesis under the guidance of a gradually diminishing representational constraint.

**Validation.** Extensive experiments demonstrate that our decoupling strategy yields substantial benefits. ERW achieves up to an $11.5\times$ training speedup to reach a comparable FID score in **350 epochs** compared to single-phase methods like REPA while achieving **FID = 1.41**. The warmup phase requires only a fraction of the total training cost, making our approach highly practical for real-world applications.

**Our contributions are threefold:**

(a) We formalize the optimization entanglement in generative models as of semantic understanding and visual synthesis, and propose a conceptual decomposition of the diffusion model into functionally specialized L2R and R2G circuits.

(b) We introduce ERW, a principled two-phase training paradigm that operationalizes this decoupling, first building a semantic foundation and then focusing on guided synthesis.

(c) We demonstrate the effectiveness of our framework through extensive experiments, achieving state-of-the-art results.

## 2 RELATED WORK

Our work builds on three research pillars: leveraging pretrained encoders, recent advances in diffusion model acceleration, and enhancing the internal representations of diffusion models through decoupled training strategies.

**Leveraging pretrained encoders for guidance.** The idea of leveraging powerful pretrained encoders (Radford et al., 2021; Oquab et al., 2023) to guide generation is well-established, with applications as GAN discriminators (Sauer et al., 2021; Kumari et al., 2022) or for knowledge distillation (Li et al., 2023b). A recent and direct approach is concurrent representation alignment, epitomized by REPA (Yu et al., 2024), which accelerates training by enforcing alignment throughout the entire process. In contrast, our work treats alignment as a foundational warmup, relaxing the constraint during later stages to allow the model to focus fully on synthesis.

**Contemporary acceleration strategies and recent advances.** Accelerating diffusion models has emerged as a critical research thrust, as recent years have witnessed significant breakthroughs across multiple fronts (Fuest et al., 2024). Post-training sampling acceleration continues to be actively pursued through knowledge distillation techniques that compress slow teachers into fast students (Salimans & Ho, 2022; Sauer et al., 2023; Shao et al., 2023), and through consistency models enabling one-shot or few-shot generation (Song et al., 2023; Heek et al., 2024). Recent work includes speculative decoding approaches for autoregressive text-to-image generation and training-free acceleration methods. Advanced numerical solvers remain crucial, with improvements to DPM-Solver (Lu et al., 2022) and novel exponential integrators significantly reducing function evaluations. Training acceleration strategies include architectural decoupling in staged pipelines (Karras et al., 2018; Ho et al., 2022; Saharia et al., 2022), curriculum learning on timesteps (Xu et al., 2024), and progressive sparse low-rank adaptation methods. ERW contributes to this rapidly evolving landscape by fundamentally decoupling learning objectives within the training process, separating semantic understanding ("what") from synthesis capability ("how").

**Internal vs. injected representations and efficient fine-tuning.** Numerous studies confirm that diffusion models implicitly learn powerful, classifier-like semantic features (Yang & Wang, 2023; Li et al., 2023a; Xiang et al., 2023), a phenomenon some works have deconstructed this phenomenon for self-supervised learning (Chen et al., 2024). An alternative strategy enhances internal representation learning by fusing diffusion objectives with auxiliary self-supervision losses, exemplified by MAGE (Li et al., 2023c) and MaskDiT (Zheng et al., 2024), which draw inspiration from contrastive learning (Chen et al., 2020a; He et al., 2020) and masked autoencoding (He et al., 2022). However, these approaches require careful balancing of competing objectives. ERW sidesteps these complexities by directly injecting mature semantic priors via dedicated warmup, freeing the model to focus purely on high-fidelity synthesis while achieving efficiency comparable to contemporary methods.

## 3 FROM FUNCTIONAL SPECIALIZATION TO DECOUPLED TRAINING IN LATENT DIFFUSION

In this section, we adopt a *three-stage* view of latent diffusion—***Pixel-to-Latent (P2L)***, ***Latent-to-Representation (L2R)***, and ***Representation-to-Generation (R2G)***— as a functional perspective that facilitates decoupled training. P2L provides compressed latents as a precondition, while L2R and R2G capture the predominant (but not exclusive) roles of early and late layers in semantic processing and generative refinement. The separation is heuristic and approximateroles overlap and are not strictly orthogonalbut it is sufficient to decouple training objectives in practice. This view underpins our two-phase framework.

### 3.1 PRELIMINARIES

**Latent diffusion models.** While classic diffusion models such as DDPM (Ho et al., 2020) adopt a discrete-time denoising process, *flow-based methods* (Lipman et al., 2022; Albergo et al., 2023; Shi et al., 2024) explore diffusion in a continuous-time setting. In particular, Scalable Interpolant Transformers (SiT) (Ma et al., 2024; Esser et al., 2024; Lipman et al., 2022; Liu et al., 2023) offer a unifying framework for training diffusion models on a continuous-time stochastic interpolant. Below, we describe how SiT can be leveraged to learn powerful latent diffusion models.

**Forward process via stochastic interpolants.** Consider a data sample $\mathbf{x} \sim p(\mathbf{x})$ (e.g., an image) and let the encoder $\mathcal{H}_{\boldsymbol{\theta}}(\mathbf{x})$ map it to its latent representation denoted as $\mathbf{z}_0 \in \mathcal{Z}$. Given standard Gaussian noise $\boldsymbol{\epsilon} \sim \mathcal{N}(\mathbf{0}, \mathbf{I})$, SiT defines a *forward process* in the latent space, parameterized by continuous time $t \in [0, 1]$:

$$\mathbf{z}_t = \alpha_t \mathbf{z}_0 + \sigma_t \boldsymbol{\epsilon} \,, \tag{1}$$

where $\alpha_t$ and $\sigma_t$ are deterministic, differentiable functions satisfying the boundary conditions:

$$(\alpha_0, \sigma_0) = (1, 0) \quad \text{and} \quad (\alpha_1, \sigma_1) = (0, 1) \,. \tag{2}$$

This construction implies that at $t = 0$ we recover the clean latent $\mathbf{z}_0$, and at $t = 1$ we have pure noise $\mathbf{z}_1 = \boldsymbol{\epsilon}$. Under mild conditions (Albergo et al., 2023), the sequence $\{\mathbf{z}_t\}$ forms a *stochastic interpolant* that smoothly transitions between data and noise in the latent space.

**Velocity-based learning.** To train a diffusion model in this continuous-time framework, SiT employs a *velocity* formulation. Differentiating $\mathbf{z}_t$ with respect to $t$ yields:

$$\dot{\mathbf{z}}_t = \dot{\alpha}_t \mathbf{z}_0 + \dot{\sigma}_t \boldsymbol{\epsilon} \,. \tag{3}$$

Conditioning on $\mathbf{z}_t$, we can rewrite the derivative as a velocity field:

$$\dot{\mathbf{z}}_t = \mathbf{F}(\mathbf{z}_t, t) \,, \tag{4}$$

where $\mathbf{F}(\mathbf{z}_t, t)$ is defined as the conditional expectation of $\dot{\mathbf{z}}_t$ given $\mathbf{z}_t$. A neural network $\mathbf{F}_{\boldsymbol{\theta}}(\mathbf{z}, t)$ is then trained to approximate $\mathbf{F}(\mathbf{z}, t)$ by minimizing:

$$\mathcal{L}_{\text{diffusion}}(\boldsymbol{\theta}) = \mathbb{E}_{\mathbf{z}_0, \boldsymbol{\epsilon}, t} \left[ \left\| \mathbf{F}_{\boldsymbol{\theta}}(\mathbf{z}_t, t) - \left( \dot{\alpha}_t \mathbf{z}_0 + \dot{\sigma}_t \boldsymbol{\epsilon} \right) \right\|^2 \right] \,. \tag{5}$$

Learning $\mathbf{F}_{\boldsymbol{\theta}}(\mathbf{z}, t)$ enables one to integrate the reverse-time ordinary differential equation (ODE) (Song et al., 2020), thereby mapping noise samples back to coherent latent representations.

## 3.2 A FUNCTIONAL CIRCUIT PERSPECTIVE FOR DECOUPLED TRAINING

Recent studies indicate that diffusion models jointly perform both *representation learning* and *generative decoding* during the denoising procedure (Yu et al., 2024; Xiang et al., 2023). Notably, every layer in the network contributes to feature extraction and generative tasks to varying degrees. To make this dual functionality clearer, we propose decomposing the diffusion process into three distinct stages: *Pixel-to-Latent* (**P2L**), *Latent-to-Representation* (**L2R**), and *Representation-to-Generation* (**R2G**), as illustrated in Figure 2. Formally, we posit that the diffusion sampling procedure can be written as:

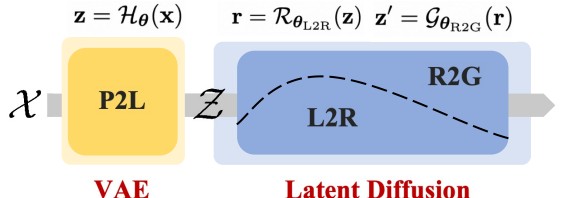

Figure 2: **Functional circuit for decoupled training.** **P2L** provides a compression precondition; early layers predominantly serve *Latent-to-Representation* (**L2R**, semantic inference), and later layers predominantly serve *Representation-to-Generation* (**R2G**, synthesis). Roles overlap in practice; we employ this perspective to organize objectives and reduce optimization entanglement.

$$\mathbf{z} = \mathcal{H}_{\boldsymbol{\theta}}(\mathbf{x}) \,, \qquad\qquad\qquad \textit{(Pixel to Latent (P2L))}$$
$$\mathbf{r} = \mathcal{R}_{\boldsymbol{\theta}_{\text{L2R}}}(\mathbf{z}) \,, \qquad\qquad\qquad \textit{(Latent to Representation (L2R))}$$
$$\mathbf{z}' = \mathcal{G}_{\boldsymbol{\theta}_{\text{R2G}}}(\mathbf{r}) \,, \qquad\qquad\qquad \textit{(Representation to Generation (R2G))}$$

Here, $\mathcal{H}_{\boldsymbol{\theta}}$ is a VAE encoder that compresses pixels to latents; $\mathcal{R}_{\boldsymbol{\theta}_{\text{L2R}}}$ and $\mathcal{G}_{\boldsymbol{\theta}_{\text{R2G}}}$ are two overlapping functional roles implemented within the shared diffusion backbone. A VAE decoder $\mathcal{D}_{\boldsymbol{\theta}}$ maps refined latents back to pixels at the end.

**Loss function decomposition.** Grounded in the augmented probability view, Appendix B (Thm. 1) gives an *exact* decomposition of the joint conditional score:

$$\nabla_{\mathbf{z}_t} \log p(\mathbf{z}_0, \mathbf{r} \mid \mathbf{z}_t, t) = \underbrace{\nabla_{\mathbf{z}_t} \log p(\mathbf{z}_0 \mid \mathbf{z}_t, \mathbf{r}, t)}_{\text{Conditional Generation Score}} + \underbrace{\nabla_{\mathbf{z}_t} \log p(\mathbf{r} \mid \mathbf{z}_t, t)}_{\text{Representation Inference Score}} \,. \tag{6}$$

This provides a principled rationale for separating optimization into representation inference (L2R) and conditional generation (R2G). In practice, we shape these two components using surrogate losses: the standard diffusion objective in Eq. (5) for generation and the alignment objective in Eq. (14) for representation; see Appendix B for details.

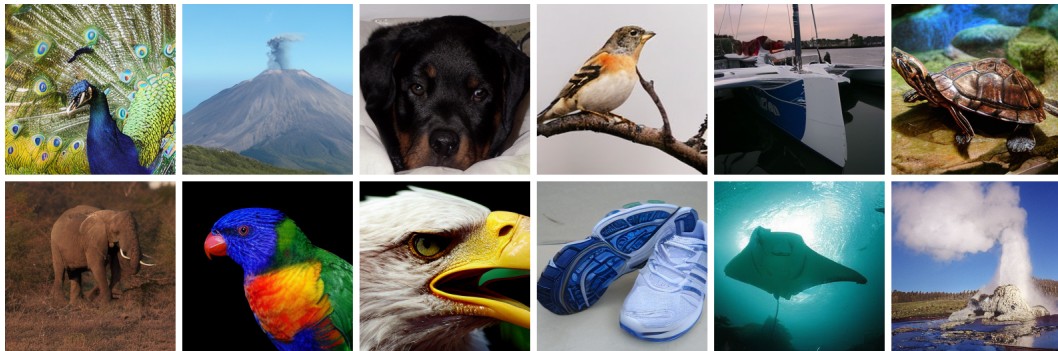

Figure 3: **Selected Samples on ImageNet** $256 \times 256$. Images generated by the SiT-XL/2 + REPA + ERW model using Classifier-Free Guidance (CFG) with a scale of $w = 1.62$ under 350 epochs.

**Stage I: Pixel-to-Latent (P2L).**    Before performing the denoising process in the high-dimensional pixel domain—where noise may obscure semantic cues—many methods Saharia et al. (2022); Ho et al. (2020); Dhariwal & Nichol (2021) compress images into a more tractable latent space:

$$\mathbf{z} = \mathcal{H}_{\boldsymbol{\theta}}(\mathbf{x}), \tag{7}$$

where $(\mathcal{H}_{\boldsymbol{\theta}}, \mathcal{D}_{\boldsymbol{\theta}})$ typically refers to a variational autoencoder or a related autoencoding architecture. This **P2L** stage reduces computational complexity and filters out low-level details, thus preserving more essential semantic information. From the perspective of the decomposed loss, P2L transforms the high-dimensional denoising problem into a lower-dimensional one where representation components (capturing semantic concepts) and reconstruction components (handling fine details) become more clearly separable, facilitating favorable conditions for separating the training stages.

**Stage II: Latent-to-Representation (L2R).**    Given a noisy latent $\mathbf{z}_t$ from the forward process (Eq. (1)), the model initially extracts a semantic representation $\mathbf{r}_t$ using the mapping $\mathcal{R}_{\boldsymbol{\theta}_{\mathrm{L2R}}}$.

$$\mathbf{r}_t = \mathcal{R}_{\boldsymbol{\theta}_{\mathrm{L2R}}}(\mathbf{z}_t, t). \tag{8}$$

This step corresponds to the *Representation Inference Score*, i.e., estimating $\nabla_{\mathbf{z}_t} \log p(\mathbf{r} \mid \mathbf{z}_t, t)$ in the augmented conditional view (Thm. 1; see also Appendix B ). Intuitively, the model should discern salient patterns (e.g., object shapes, style characteristics, or conditioning signals) before denoising. Under the sufficient statistic assumption (see Assump. 1), the representation $\mathbf{r}_t$ effectively captures the essential information from the latent $\mathbf{z}_t$. The true representation score available one could consider the idealized regression objective

$$\min_{\mathcal{R}} \ \mathbb{E}_{t,\mathbf{z}_t} \left[ \| \mathcal{R}_{\boldsymbol{\theta}_{\mathrm{L2R}}}(\mathbf{z}_t, t) - \nabla_{\mathbf{z}_t} \log p(\mathbf{r} \mid \mathbf{z}_t, t) \|^2 \right]. \tag{9}$$

In practice, we do not access this score; instead we employ surrogate alignment losses: the clean-latent warmup in Eq. (13) and the noisy-input alignment term in Eq. (14). By explicitly decoupling the objective for semantic feature extraction from that of generative refinement, the model is guided to learn representations and ensures that the early layers focus on capturing semantic features.

**Stage III: Representation-to-Generation (R2G).**    In the final phase of each reverse diffusion update in (3), known as the R2G stage, the extracted semantic representation is transformed into an updated latent with reduced noise:

$$\mathbf{z}_{t-\Delta t} = \mathcal{G}_{\boldsymbol{\theta}_{\mathrm{R2G}}}(\mathbf{r}_t, t). \tag{10}$$

This output serves the same purpose as the $\mathbf{z}'$ term introduced, but is specifically defined for the discrete time step $t - \Delta t$ in the continuous-time diffusion process. In the decomposition, this step aligns with the *Conditional Generation Score* component. For the rigorous joint-conditional view, see Thm. 1. The conditional generation score available one could consider the idealized regression objective

$$\min_{\mathcal{G}} \ \mathbb{E}_{t,\mathbf{z}_t} \left[ \| \mathcal{G}_{\boldsymbol{\theta}_{\mathrm{R2G}}}(\mathbf{r}_t, t) - \nabla_{\mathbf{z}_t} \log p(\mathbf{z}_0 \mid \mathbf{z}_t, \mathbf{r}_t, t) \|^2 \right]. \tag{11}$$

In practice, we instead rely on the standard diffusion objective in Eq. (5) (and its Phase 2 combination in Eq. (14)) to shape the generation component while using the learned representations as

guidance. Injecting the semantic representation $\mathbf{r}_t$ into a cleaner latent $\mathbf{z}_{t-\Delta t}$ which is significantly less noisy than $\mathbf{z}_t$ ensures that abstract semantic features are effectively transformed into the precise latent elements required for content generation. Meanwhile, the cross interaction between L2R and R2G (also discussed in Appendix B) is empirically small when the two gradients are sufficiently separated in function, helping to mitigate destructive interference.

**Two-stage sampling process: Representation extraction precedes generation.** Within the continuous-time framework, after mapping pixel data to the latent space through P2L, each infinitesimally small time step during the reverse SDE update can be interpreted as a two-stage process:

$$\mathbf{r}_t = \mathcal{R}_{\boldsymbol{\theta}_{\mathrm{L2R}}}(\mathbf{z}_t, t) \quad \longrightarrow \quad \mathbf{z}_{t-\Delta t} = \mathcal{G}_{\boldsymbol{\theta}_{\mathrm{R2G}}}(\mathbf{r}_t, t) \tag{12}$$

Hence, every time step naturally splits into **(i)** L2R for refining the representation and **(ii)** R2G for synthesizing an updated latent. This loop neatly implements the principle of "first representation, then generation". Empirically, prior work (Yu et al., 2024; Xiang et al., 2023) confirms that early layers of the diffusion model predominantly focus on representation extraction, whereas later layers emphasize generative refinement. Consequently, the staged design mirrors the reverse-time diffusion trajectory, concluding in a final latent $\mathbf{z}_0$ that is decoded via $\mathcal{D}_{\boldsymbol{\theta}}$ to yield the synthesized output $\mathbf{x}_0$.

### 3.3 Embedded Representation Warmup: Training with Two Phases

Guided by the circuit view and our augmented-space analysis, we present Embedded Representation Warmup **(ERW)**, a framework that strategically decouples training into two phases. In **Phase 1**, we initialize the early layers of the diffusion model with high-quality semantic features from pretrained models; in **Phase 2**, we transition to standard diffusion training with a gradually diminishing representation alignment term, allowing the model to increasingly focus on generation. This mirrors the sampling loop: first infer representation, then generate.

**Phase 1: Representation Warmup Stage** To alleviate the burden of learning semantic features from scratch, we begin with a dedicated warmup stage. During this phase, the model's L2R circuit is initialized to align with semantically rich features extracted from a pretrained representation model (e.g., DINOv2, MAE, or CLIP). Let $\mathcal{H}_{\boldsymbol{\theta}}(\mathbf{x})$ denote an encoder that maps an image $\mathbf{x} \in \mathcal{X}$ to its latent representation $\mathbf{z} \in \mathcal{Z}$, and let $\mathbf{f}_{\mathrm{rep}} : \mathcal{X} \to \mathcal{R}$ be a high-quality pretrained representation model. We use a single alignment objective shared by both phases:

$$\mathcal{L}_{\mathrm{align}}(k) = \mathbb{E}_{\mathbf{x}, \boldsymbol{\epsilon}, t}\Big[ s(k, t)\, \ell_{\mathrm{NT\text{-}Xent}}\Big( \mathcal{T}_{\boldsymbol{\theta}}(\mathcal{R}_{\boldsymbol{\theta}_{\mathrm{L2R}}}(\mathbf{z}_t, t)),\ \mathbf{f}_{\mathrm{rep}}(\mathbf{x}) \Big) \Big]. \tag{13}$$

Here $\mathbf{z}_t$ and the schedule $s(k, t)$ are

$$\mathbf{z}_t = \begin{cases} \mathcal{H}_{\boldsymbol{\theta}}(\mathbf{x}) & (t{=}0) \\ \alpha_t\, \mathbf{z}_0 + \sigma_t\, \boldsymbol{\epsilon} & (t{>}0),\ \mathbf{z}_0 = \mathcal{H}_{\boldsymbol{\theta}}(\mathbf{x}) \end{cases} \quad \text{and} \quad s(k, t) = \begin{cases} 1 & (t{=}0) \\ \lambda_{\mathrm{train}}(k) = c_0\, \exp\!\big(-\tfrac{k}{\tau}\big) & (t{>}0) \end{cases}.$$

Warmup sets $t{=}0$; Phase 2 samples $t \sim \mathcal{U}[0, 1]$ and uses the decayed $\lambda_{\mathrm{train}}(k) = c_0\, \exp\!\big(-k/\tau\big)$ to gradually shift focus from alignment to generation, where $k$ is the training step, and $c_0, \tau$ are hyperparameters.

**Phase 2: Generative Training with Decaying Representation Guidance** After the warmup stage has effectively initialized the diffusion model with semantically rich features, we proceed with a joint objective that combines the standard diffusion loss with a gradually diminishing representation alignment term. Formally, the overall training loss is given by:

$$\mathcal{L}_{\mathrm{total}} = \mathcal{L}_{\mathrm{diffusion}} + \lambda_{\mathrm{train}}(k) \cdot \mathcal{L}_{\mathrm{align}} \tag{14}$$

Here, $\mathcal{L}_{\mathrm{diffusion}}$ denotes the velocity prediction loss as defined in Eq. (5), and the alignment term is the objective in Eq. (13). The weight $\lambda_{\mathrm{train}}(k)$ modulates the impact of alignment during training. In practice, we instantiate $\ell_{\mathrm{NT\text{-}Xent}}$ with in-batch negatives and use the same projection head $\mathcal{T}_{\boldsymbol{\theta}}$ across both phases (Section 4.1). The alignment thus acts as a weak semantic tether late in training, mitigating forgetting while letting R2G dominate. Both phases share the same alignment loss $\ell_{\mathrm{NT\text{-}Xent}}\big(\mathcal{T}_{\boldsymbol{\theta}}(\cdot), \mathbf{f}_{\mathrm{rep}}(\cdot)\big)$; they differ only in (i) the noise level of the input (clean $t{=}0$ in Phase 1 vs. noisy $t{>}0$ in Phase 2) and (ii) the schedule $\lambda_{\mathrm{train}}(k)$ (absent in warmup, exponentially decayed in Phase 2). Consistent with the augmented-space identity (Thm. 1), the surrogate gradient decomposes as

$$\nabla_{\boldsymbol{\theta}} \mathcal{L}_{\mathrm{total}}(k) \approx \mathbb{E}\Big[ \underbrace{\nabla_{\boldsymbol{\theta}} \mathcal{L}_{\mathrm{diffusion}}}_{\text{shapes} \nabla_{\mathbf{z}_t} \log p(\mathbf{z}_0 | \mathbf{z}_t, \mathbf{r}, t)} + s(k, t)\, \underbrace{\nabla_{\boldsymbol{\theta}} \mathcal{L}_{\mathrm{align}}(k)}_{\text{shapes} \nabla_{\mathbf{z}_t} \log p(\mathbf{r} | \mathbf{z}_t, t)} \Big],$$

up to standard surrogate mismatches. This makes the training-time decomposition mirror the sampling-time loop: first representation (L2R), then generation (R2G).

# 4 EXPERIMENTS

In this section, we provide a comprehensive evaluation of our proposed ERW approach. We begin by outlining experimental setups ( Section 4.1 ), including dataset and implementation details. Next, we present comparisons with state-of-the-art baselines to demonstrate the benefits of ERW in both FID and training speed ( Section 4.2 ). We then analyze the role of our warmup procedure in boosting training efficiency ( Section 4.3 ). Finally, we conduct ablation studies to examine the effects of various alignment strategies, architecture depths, and target representation models ( Section 4.4 ).

## 4.1 SETUP

• **Implementation Details.** We adhere closely to the experimental setups described in DiT (Peebles & Xie, 2023) and SiT (Ma et al., 2024), un-

Table 1: **System-level comparison** on ImageNet $256 \times 256$ with CFG. ↓ and ↑ indicate whether lower or higher values are better, respectively. Results marked with an asterisk (*) use advanced CFG scheduling techniques; specifically, for our method, we apply the guidance interval scheduling from (Kynkäänniemi et al., 2024).

| Model | Epochs | FID↓ | sFID↓ | IS↑ | Pre.↑ | Rec.↑ |
|---|---|---|---|---|---|---|
| *Pixel diffusion* | | | | | | |
| ADM-U | 400 | 3.94 | 6.14 | 186.7 | 0.82 | 0.52 |
| VDM++ | 560 | 2.40 | - | 225.3 | - | - |
| Simple diffusion | 800 | 2.77 | - | 211.8 | - | - |
| CDM | 2160 | 4.88 | - | 158.7 | - | - |
| *Latent diffusion, U-Net* | | | | | | |
| LDM-4 | 200 | 3.60 | - | 247.7 | 0.87 | 0.48 |
| *Latent diffusion, Transformer + U-Net hybrid* | | | | | | |
| U-ViT-H/2 | 240 | 2.29 | 5.68 | 263.9 | 0.82 | 0.57 |
| DiffiT* | - | 1.73 | - | 276.5 | 0.80 | 0.62 |
| MDTv2-XL/2* | 1080 | 1.58 | 4.52 | 314.7 | 0.79 | 0.65 |
| *Latent diffusion, Transformer* | | | | | | |
| MaskDiT | 1600 | 2.28 | 5.67 | 276.6 | 0.80 | 0.61 |
| SD-DiT | 480 | 3.23 | - | - | - | - |
| DiT-XL/2 | 1400 | 2.27 | 4.60 | 278.2 | **0.83** | 0.57 |
| SiT-XL/2 | 1400 | 2.06 | 4.50 | 270.3 | 0.82 | 0.59 |
| + REPA | 200 | 1.96 | 4.49 | 264.0 | 0.82 | 0.60 |
| + REPA* | 800 | 1.42 | 4.70 | 305.7 | 0.80 | 0.65 |
| + ERW (ours) | 200 | 1.64 | 4.71 | 260.2 | 0.78 | **0.66** |
| **+ ERW (ours)*** | **350** | **1.41** | **4.46** | **293.9** | 0.79 | 0.65 |

less otherwise noted. Specifically, we utilize the ImageNet dataset (Deng et al., 2009), preprocessing each image to a resolution of $256 \times 256$ pixels. Following the protocols of ADM (Dhariwal & Nichol, 2021), each image is encoded into a compressed latent vector $\mathbf{z} \in \mathbb{R}^{32 \times 32 \times 4}$ using the Stable Diffusion VAE (Rombach et al., 2022). For our model configurations, we employ the B/2 and XL/2 architectures as introduced in the SiT papers, which process inputs with a patch size of 2. To ensure a fair comparison with SiT models and REPA, we maintain a consistent batch size of 256 throughout training. Further experimental details, including hyperparameter settings and computational resources, are provided in  Appendix D .

• **Evaluation.** We report Fréchet inception distance (FID; Heusel et al. 2017), sFID (Nash et al., 2021), inception score (IS; Salimans et al. 2016), precision (Pre.) and recall (Rec.) (Kynkäänniemi et al., 2019) using 50K samples. We also include CKNNA (Huh et al., 2024) as discussed in ablation studies. Detailed setups for evaluation metrics are provided in  Appendix E .

• **Sampler and Alignment objective.** Following SiT (Ma et al., 2024), we always use the SDE Euler-Maruyama sampler (for SDE with $w_t = \sigma_t$) and set the number of function evaluations (NFE) as 250 by default. We use Normalized Temperature-scaled Cross Entropy (NT-Xent) training objective for alignment.

• **Baselines.** We use several recent diffusion-based generation methods as baselines, each employing different inputs and network architectures. Specifically, we consider the following four types of approaches: (a) *Pixel diffusion*: ADM (Dhariwal & Nichol, 2021), VDM++ (Kingma & Gao, 2024), Simple diffusion (Hoogeboom et al., 2023), CDM (Ho et al., 2022), (b) *Latent diffusion with U-Net*: LDM (Rombach et al., 2022), (c) *Latent diffusion with transformer+U-Net hybrid models*: U-ViT-H/2 (Bao et al., 2023), DiffiT (Hatamizadeh et al., 2024), and MDTv2-

Table 2: **FID comparisons with SiT-XL/2.** In this table, we report the FID of ERW with SiT-XL/2 on ImageNet $256 \times 256$ at various Training iterations. Here is only full training without warmup, because we load a well trained warmuped checkpoint. For comparison, we also present the performance of the state-of-the-art baseline REPA at similar iterations or comparable FID values. Note that ↓ indicates that lower values are preferred and all results reported are without Classifier-Free Guidance.

| Model | #Params | Iter. | FID↓ | sFID↓ | IS↑ | Prec.↑ | Rec.↑ |
|---|---|---|---|---|---|---|---|
| SiT-XL/2 | 675M | 7M | 8.3 | 6.32 | 131.7 | 0.68 | 0.67 |
| +REPA | 675M | 50K | 52.3 | 31.24 | 24.3 | 0.45 | 0.53 |
| +ERW (ours) | 675M | 50K | 25.0 | 12.06 | 56.1 | 0.62 | 0.57 |
| +REPA | 675M | 100K | 19.4 | 6.06 | 67.4 | 0.64 | 0.61 |
| +ERW (ours) | 675M | 100K | 12.1 | 5.25 | 94.2 | 0.69 | 0.63 |

XL/2 (Gao et al., 2023), and (d) *Latent diffusion with transformers*: MaskDiT (Zheng et al., 2024), SD-DiT (Zhu et al., 2024), DiT (Peebles & Xie, 2023), and SiT (Ma et al., 2024). Here, we refer to Transformer+U-Net hybrid models that contain skip connections, which are not originally used in pure transformer architecture. Details are provided in  Appendix F .

## 4.2 COMPARISON

Table 1 summarizes our results on ImageNet $256 \times 256$ under Classifier-Free Guidance (CFG). Our ERW significantly boosts the convergence speed of SiT-XL/2, enabling strong FID scores at just 350 epochs. As shown in Table 1, our method achieves an FID of **1.41** in **350 epochs**, that REPA requires 800 epochs to approach, demonstrating a high speedup while achieving state-of-the-art performance. Figure 3 illustrates generated samples, further confirming the high-quality outputs achieved by ERW.

## 4.3 ERW EFFICIENCY

We begin by how ERW influences SiT-XL/2's FID when w/o CFG.

- **Efficient FID Improvements.** In Table 2, ERW consistently achieves competitive or superior FID values compared to baselines.

Table 3: **Analysis of ERW depth, projection depth, and different dynamic or consistent projection loss $\lambda$ influences in SiT-XL/2.** All models are based on SiT-XL/2 and trained for 100K iterations under a batch size of 256 without using Classifier-Free Guidance on ImageNet $256 \times 256$. The target representation model is DINOv2-B, and the objective is NT-Xent. $\downarrow$ indicates lower values are better. The results show that a projection depth of 14 and a projection loss $\lambda$ of 4.0 yield substantial improvements in both FID and sFID, indicating an optimal configuration for model performance.

| ERW Depth | Proj. Depth | $\lambda$ | FID↓ | sFID↓ | IS↑ | Prec.↑ | Rec.↑ |
|---|---|---|---|---|---|---|---|
| SiT-XL/2 + REPA (Yu et al., 2024) | | | 19.4 | 6.06 | 67.4 | 0.64 | 0.61 |
| 3 | 8 | 0.5 | 14.4 | **5.28** | 82.7 | 0.68 | 0.62 |
| 4 | 8 | 0.5 | 13.8 | 5.31 | 87.1 | 0.68 | 0.62 |
| 5 | 8 | 0.5 | 13.4 | 5.29 | 87.8 | 0.68 | 0.63 |
| 6 | 8 | 0.5 | 13.6 | 5.30 | 87.3 | 0.67 | 0.63 |
| 8 | 8 | 0.5 | 15.4 | 5.37 | 82.3 | 0.66 | 0.63 |
| 12 | 8 | 0.5 | 16.2 | 5.36 | 79.2 | 0.66 | 0.63 |
| 5 | 10 | 0.5 | 12.9 | 5.29 | 90.4 | 0.68 | 0.63 |
| 5 | 12 | 0.5 | 12.5 | 5.24 | 92.0 | 0.69 | 0.62 |
| 5 | 14 | 0.5 | 12.5 | 5.26 | 91.5 | 0.69 | 0.63 |
| 5 | 16 | 0.5 | 12.3 | 5.25 | 93.4 | 0.69 | 0.62 |
| 5 | 18 | 0.5 | **12.1** | 5.25 | **94.2** | **0.69** | 0.63 |
| 5 | 20 | 0.5 | 12.6 | 5.27 | 92.3 | 0.69 | 0.63 |
| 5 | 18 | 0.1 | 16.6 | 5.31 | 75.8 | 0.67 | 0.60 |
| 5 | 18 | 1.0 | 12.7 | 5.41 | 92.8 | 0.68 | 0.64 |
| 5 | 18 | 2.0 | 13.3 | 5.39 | 90.5 | 0.68 | 0.63 |
| 5 | 18 | 4.0 | 13.1 | 5.38 | 92.2 | 0.68 | **0.64** |
| 5 | 18 | 6.0 | 13.4 | 5.45 | 91.6 | 0.67 | 0.63 |

For instance, ERW reaches an FID of 12.1 with 100k warmup + 100k full training, markedly outperforming the REPA method (Yu et al., 2024) which scores 19.4 within the same budget.

- **Leveraging Pretrained Features.** This gain highlights the advantage of injecting pretrained semantic priors via warmup, thereby accelerating the full training.

**Warmup versus full training.** Next, we analyze how splitting the total training budget between warmup and full diffusion training impacts both generation quality and computational overhead. As shown in Figure 4, the FLOPs for the warmup phase are significantly lower than for the full training phase.

Table 4: **Analysis of ERW** on ImageNet $256 \times 256$. All models are SiT-B/2 trained for 50K iterations. All metrics except FID without Classifier-Free Guidance. We fix $\lambda = 0.5$ here. $\downarrow$ and $\uparrow$ indicate whether lower or higher values are better, respectively.

| Target Repr. | Depth | Objective | FID↓ | sFID↓ | IS↑ | Prec.↑ | Rec.↑ |
|---|---|---|---|---|---|---|---|
| MoCov3-B | 8 | NT-Xent | 61.1 | **7.6** | 22.38 | 0.42 | **0.58** |
| MoCov3-L | 8 | NT-Xent | 73.0 | 8.0 | 17.96 | 0.38 | 0.52 |
| CLIP-L | 8 | NT-Xent | 58.9 | 7.7 | 23.68 | **0.44** | 0.54 |
| DINOv2-B | 8 | NT-Xent | 55.6 | 7.8 | 25.45 | **0.44** | 0.56 |
| DINOv2-L | 8 | NT-Xent | **55.5** | 7.8 | 25.45 | **0.44** | 0.56 |
| DINOv2-g | 8 | NT-Xent | 59.4 | **7.6** | **25.53** | **0.44** | 0.56 |

## 4.4 ABLATION STUDIES

We further dissect the effectiveness of ERW by conducting ablation studies on various design choices and parameter settings.

**Target representation.** We first compare alignment with multiple self-supervised encoders: MoCov3, CLIP, and DINOv2, as summarized in Table 4.

- **Universality of Pretrained Encoders.** All encoders tested offer improvements over baselines, indicating that ERW can benefit from a range of representation models.

Table 5: **Analysis of ERW places influences in SiT-B/2.** All models are based on SiT-B/2 and trained for 50K iterations under the batch size of 256 without using Classifier-Free Guidance on ImageNet $256 \times 256$. $\downarrow$ indicates lower values are better. Results empirically validate our hypothesis that placing ERW at the forefront of the architecture yields optimal performance.

| Target Repr. | Depth | Objective | FID↓ | sFID↓ | IS↑ | Prec.↑ | Rec.↑ |
|---|---|---|---|---|---|---|---|
| SiT-B/2 + REPA (Yu et al., 2024) | | | 78.2 | 11.71 | 17.1 | 0.33 | 0.48 |
| DINOv2-B | 0-8 | NT-Xent | **54.2** | **8.12** | **27.2** | **0.45** | **0.59** |
| DINOv2-B | 1-9 | NT-Xent | 69.1 | 13.0 | 18.7 | 0.37 | 0.51 |
| DINOv2-B | 2-10 | NT-Xent | 67.7 | 13.4 | 19.0 | 0.38 | 0.52 |
| DINOv2-B | 3-11 | NT-Xent | 67.5 | 11.8 | 19.5 | 0.38 | 0.52 |
| DINOv2-B | 4-11 | NT-Xent | 67.8 | 13.1 | 19.0 | 0.38 | 0.52 |

provements over baselines, indicating that ERW can benefit from a range of representation models.

- **Marginal Differences among DINOv2 Variants.** DINOv2-B, DINOv2-L, and DINOv2-g yield comparable gains, suggesting that ERW does not require the largest possible teacher encoder for effective representation transfer. This suggests that ERW is not limited to a specific encoder architecture but can leverage a wide range of powerful, pretrained feature extractors, making it a versatile tool for accelerating diffusion model training.

**Placement of ERW Depth.** We hypothesize that early layers in the diffusion backbone primarily learn semantic features (the L2R circuit), whereas deeper layers specialize in generative decoding. The placement of the alignment loss is therefore critical. We specify the alignment target using "Depth X-Y", which means the alignment loss is computed on the output of layer Y, using a projection head that takes features from layers X through Y as input.

• *Empirical Validation.* In Table 5 , initializing the earliest layers (0–8) notably outperforms re-initializing middle or late sections (FID 54.2 vs. > 67).

• *Consistent with Circuit Perspective.* This corroborates our three-stage diffusion circuit ( Section 3 ), underscoring that aligning deeper layers for representation can be suboptimal since those layers focus on generation. Targeting the initial layers for warmup is therefore crucial, reinforcing our theoretical claim that representation learning is predominantly the function of the early network stages, while later stages are specialized for generative refinement.

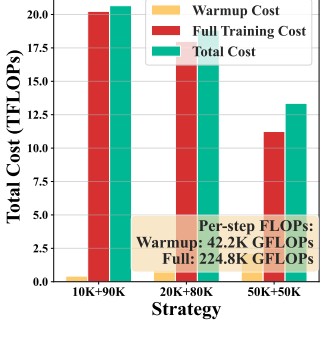

Figure 4: **Comparison of Training Efficiency and Cost Analysis with Warmup and Full Training Stages.** Bar chart comparing the computational costs of the warmup and full training stages for different strategies. The chart shows the warmup cost, full training cost, and their corresponding total cost.

**Projection depth and alignment weight.** We also investigate how the final projection head depth and the alignment-loss coefficient $\lambda$ affect training ( Table 3 ). The projection head, $\mathcal{T}_\theta$, is a deep MLP that maps the features to the dimensionality of the target representation as same as REPA.

• *Empirical Validation.* Using 5 warmup layers, a projection head at depth 18, and $\lambda = 0.5$ achieves an FID of **12.1** at 100k iterations—a substantial gain over baselines.

• *Trade-off in $\lambda$.* Larger $\lambda$ offers stronger representation alignment initially but may disrupt convergence if pushed to extremes, highlighting the need for moderate scheduling.

**Representation dynamics.** We examine the temporal progression of representation alignment in Figure 5 .

• *Initial Dip, Subsequent Recovery.* Alignment falls early on as the pretrained features adjust to the diffusion objectives, but it then recovers and improves.

• *Role of Decaying Guidance.* A decaying weight in the alignment term ( Section 3.3 ) fosters stable synergy between semantic alignment and generative refinement. The representation alignment thus follows a U-shaped trajectory, revealing the model's initial adaptation of pretrained features to the diffusion task, followed by a distillation into robust, generation-aligned embeddings.

**CKNNA analysis.** Finally, we measure layer-wise representation quality using Class-conditional $k$-Nearest Neighbor Accuracy (CKNNA) (Caron et al., 2021), which indicates how well the hidden features capture class discriminability.

• *Improved Semantic Alignment.* ERW yields systematically higher CKNNA scores, confirming stronger semantic preservation.

• *Evolving Layer-wise Semantics.* The alignment initially drops then recovers, mirroring the trends seen in Figure 5 and pretrained features are effectively integrated rather than merely overwritten.

Figure 5: **Scalability of ERW.** Training dynamics for alignment indicate that within the 500K training steps for SiT-XL/2, the alignment between DINOv2-g and the diffusion model first decreases and then increases.

## 5 CONCLUSION AND FUTURE WORK

In this work, we introduced Embedded Representation Warmup (ERW), a novel two-phase training framework that significantly enhances the training efficiency of diffusion models. By dedicating an initial phase to align the model's early layers with a pretrained encoder, ERW establishes a strong semantic foundation that accelerates the subsequent generative training. Our key innovation is the explicit separation of representation alignment and generation, which, when combined with a decaying alignment schedule, proves more effective than continuous, single-phase regularization. We demonstrated empirically that ERW leads to substantial speedups in training convergence up to $11.5\times$ compared to REPA and achieves FID=1.41 with 350 epochs. Our ablations confirmed that targeting the early layers is crucial and that the two-phase approach is a cost-effective strategy for high-fidelity generative modeling.

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

CONTENTS

# A    LLM USAGE STATEMENT

LLMs were used solely as auxiliary tools for grammar checking and language polishing. They did not contribute to the generation of research ideas, the design of experiments, the development of methodologies, data analysis, or any substantive aspects of the research. All scientific content, conceptual contributions, and experimental results are entirely the work of the authors. The authors take full responsibility for the contents of this paper.

# B    THEORETICAL ANALYSIS

In this section, we provide a principled theoretical foundation for our ERW. We move beyond the empirical intuition of entangled objectives and demonstrate that our approach naturally emerges from a more fundamental perspective: conditional score matching within an *augmented probability space*. This formulation recasts the generative modeling problem as one where semantic understanding is an explicit conditional variable, thereby justifying the decoupling of representation learning from the synthesis process.

## B.1    PRELIMINARIES

To ensure clarity, we first establish the key mathematical objects used in our analysis. Let $\mathcal{X}$ be the high-dimensional data space (e.g., images), $\mathcal{Z}$ be the compressed latent space from a VAE, and $\mathcal{R}$ be the semantic representation space. We work with the following variables:
- $\mathbf{x} \in \mathcal{X}$: A sample from the data distribution $p_{\text{data}}(\mathbf{x})$.
- $\mathbf{z}_0 \in \mathcal{Z}$: The clean latent representation of $\mathbf{x}$, obtained via a VAE encoder $\mathcal{H}_{\boldsymbol{\theta}_{\text{VAE}}}(\mathbf{x})$.
- $\mathbf{z}_t \in \mathcal{Z}$: The noisy latent at time $t \in [0, 1]$, defined by the forward process $\mathbf{z}_t = \alpha_t \mathbf{z}_0 + \sigma_t \boldsymbol{\epsilon}$ where $\boldsymbol{\epsilon} \sim \mathcal{N}(\mathbf{0}, \mathbf{I})$.
- $\mathbf{r} \in \mathcal{R}$: A high-level semantic representation vector corresponding to $\mathbf{x}$.

The key functions and models in our framework are:
- $\mathbf{f}_{\text{rep}} : \mathcal{X} \to \mathcal{R}$: A powerful, pretrained, and fixed representation model (e.g., DINOv2) that maps a clean image $\mathbf{x}$ to its semantic representation $\mathbf{r}$.
- $\mathbf{F}_{\boldsymbol{\theta}}(\mathbf{z}_t, t, \mathbf{r})$: The diffusion model we aim to train, which predicts the score or velocity, potentially conditioned on a semantic representation $\mathbf{r}$.
- $\mathcal{R}_{\boldsymbol{\theta}_{\text{L2R}}}$: The sub-network corresponding to the L2R circuit, which extracts representations from $\mathbf{z}_t$.
- $\mathcal{G}_{\boldsymbol{\theta}_{\text{R2G}}}$: The sub-network corresponding to the R2G circuit, which performs generation based on an extracted representation.

The goal of a diffusion model is to learn the score function $\nabla_{\mathbf{z}_t} \log p(\mathbf{z}_t)$, which guides the reverse process from noise back to data. In the standard formulation, this requires learning to denoise across all time steps $t \in [0, 1]$ without explicit semantic guidance.

## B.2    A PRINCIPLED VIEW VIA AN AUGMENTED PROBABILITY SPACE

We formalize the intuition of decoupling representation and generation by constructing an augmented probability space that explicitly includes the semantic representation $\mathbf{r}$ as a random variable. This principled view demonstrates that our two-phase training strategy naturally emerges from optimizing a conditional score-matching objective in this richer probabilistic landscape.

### B.2.1    CONSTRUCTION OF THE AUGMENTED SPACE

We define an augmented probability space over the tuple $(\mathbf{z}_0, \mathbf{z}_t, \mathbf{r}, t)$, where the joint distribution factorizes as:
$$p(\mathbf{z}_0, \mathbf{z}_t, \mathbf{r}, t) = p(\mathbf{z}_t \mid \mathbf{z}_0, t)\, p(\mathbf{r} \mid \mathbf{z}_0)\, p(\mathbf{z}_0)\, p(t) \tag{15}$$

This factorization leverages the conditional independence assumptions inherent in the diffusion process. Specifically, given the clean latent $\mathbf{z}_0$, the noisy latent $\mathbf{z}_t$ is independent of the semantic representation $\mathbf{r}$, and both are independent of the time variable $t$. Each component has a clear interpretation:

$$p(\mathbf{z}_0) = \int p_{\text{data}}(\mathbf{x}) \delta(\mathbf{z}_0 - \mathcal{H}_{\boldsymbol{\theta}_{\text{VAE}}}(\mathbf{x})) d\mathbf{x} \tag{16}$$

$$p(t) = \mathcal{U}[0, 1] \quad \text{(uniform time distribution)} \tag{17}$$

$$p(\mathbf{z}_t \mid \mathbf{z}_0, t) = \mathcal{N}(\mathbf{z}_t; \alpha_t \mathbf{z}_0, \sigma_t^2 \mathbf{I}) \tag{18}$$

$$p(\mathbf{r} \mid \mathbf{z}_0) = \delta\left(\mathbf{r} - \mathbf{f}_{\text{rep}}\left(\mathcal{D}_{\boldsymbol{\theta}_{\text{VAE}}}(\mathbf{z}_0)\right)\right) \tag{19}$$

Here, $\mathcal{D}_{\boldsymbol{\theta}_{\mathrm{VAE}}}$ denotes the VAE decoder that maps latents back to image space. The distribution $p(\mathbf{z}_0)$ in (16) represents the VAE's learned prior over clean latents, induced by the data distribution through the encoder. The forward kernel (18) follows the standard diffusion forward process with noise scheduling parameters $\alpha_t$ and $\sigma_t$.

The key insight is that equation (19) deterministically links semantic representations to clean latents through the Dirac delta function, transforming the unconditional generation problem into a semantically-conditioned one. This constraint ensures that every clean latent $\mathbf{z}_0$ has a uniquely associated semantic representation $\mathbf{r}$, creating a deterministic mapping from the latent space to the representation space.

### B.2.2 MARGINAL AND CONDITIONAL DISTRIBUTIONS

From the joint distribution, we can derive several important marginal and conditional distributions through careful integration.

**Marginal over noisy latents:** The marginal distribution over noisy latents is obtained by integrating out the semantic representation $\mathbf{r}$:

$$p(\mathbf{z}_t, t) = \int \int \int p(\mathbf{z}_0, \mathbf{z}_t, \mathbf{r}, t) \, d\mathbf{z}_0 \, d\mathbf{r} \tag{20}$$

$$= \int \int \int p(\mathbf{z}_t \mid \mathbf{z}_0, t) \, p(\mathbf{r} \mid \mathbf{z}_0) \, p(\mathbf{z}_0) \, p(t) \, d\mathbf{z}_0 \, d\mathbf{r} \tag{21}$$

$$= p(t) \int p(\mathbf{z}_0) p(\mathbf{z}_t \mid \mathbf{z}_0, t) \left( \int p(\mathbf{r} \mid \mathbf{z}_0) d\mathbf{r} \right) d\mathbf{z}_0 \tag{22}$$

$$= p(t) \int p(\mathbf{z}_0) p(\mathbf{z}_t \mid \mathbf{z}_0, t) \, d\mathbf{z}_0 \tag{23}$$

where the integral $\int p(\mathbf{r} \mid \mathbf{z}_0) d\mathbf{r} = 1$ since $p(\mathbf{r} \mid \mathbf{z}_0)$ is a valid probability distribution. This recovers the standard marginal distribution used in unconditional diffusion models.

**Joint marginal over** $(\mathbf{z}_t, \mathbf{r}, t)$**:** More critically for our analysis, we can compute the joint marginal over $(\mathbf{z}_t, \mathbf{r}, t)$ by integrating out only $\mathbf{z}_0$:

$$p(\mathbf{z}_t, \mathbf{r}, t) = \int p(\mathbf{z}_0, \mathbf{z}_t, \mathbf{r}, t) \, d\mathbf{z}_0 \tag{24}$$

$$= \int p(\mathbf{z}_t \mid \mathbf{z}_0, t) \, p(\mathbf{r} \mid \mathbf{z}_0) \, p(\mathbf{z}_0) \, p(t) \, d\mathbf{z}_0 \tag{25}$$

$$= p(t) \int p(\mathbf{z}_0) p(\mathbf{z}_t \mid \mathbf{z}_0, t) \delta \left( \mathbf{r} - \mathbf{f}_{\mathrm{rep}} \left( \mathcal{D}_{\boldsymbol{\theta}_{\mathrm{VAE}}}(\mathbf{z}_0) \right) \right) d\mathbf{z}_0 \tag{26}$$

Using the property of the Dirac delta function, this integral evaluates to:

$$p(\mathbf{z}_t, \mathbf{r}, t) = p(t) \int_{\mathbf{z}_0 : \mathbf{f}_{\mathrm{rep}}(\mathcal{D}_{\boldsymbol{\theta}_{\mathrm{VAE}}}(\mathbf{z}_0)) = \mathbf{r}} p(\mathbf{z}_0) p(\mathbf{z}_t \mid \mathbf{z}_0, t) \, d\mathbf{z}_0 \tag{27}$$

where the integration is restricted to the set of clean latents $\mathbf{z}_0$ that produce the semantic representation $\mathbf{r}$ when decoded and passed through the representation function.

**Conditional distribution for generation:** We can also derive the conditional distribution of clean latents given noisy latents and semantic representations:

$$p(\mathbf{z}_0 \mid \mathbf{z}_t, \mathbf{r}, t) = \frac{p(\mathbf{z}_0, \mathbf{z}_t, \mathbf{r}, t)}{p(\mathbf{z}_t, \mathbf{r}, t)} \tag{28}$$

$$= \frac{p(\mathbf{z}_t \mid \mathbf{z}_0, t) \, p(\mathbf{r} \mid \mathbf{z}_0) \, p(\mathbf{z}_0) \, p(t)}{p(\mathbf{z}_t, \mathbf{r}, t)} \tag{29}$$

$$= \frac{p(\mathbf{z}_t \mid \mathbf{z}_0, t) \, \delta \left( \mathbf{r} - \mathbf{f}_{\mathrm{rep}} \left( \mathcal{D}_{\boldsymbol{\theta}_{\mathrm{VAE}}}(\mathbf{z}_0) \right) \right) \, p(\mathbf{z}_0)}{p(\mathbf{z}_t, \mathbf{r}, t)} \tag{30}$$

This conditional distribution is the target that our diffusion model seeks to approximate, representing the posterior over clean latents given both the noisy observation and the semantic constraint.

The key insight from equation (27) is that the semantic constraint creates a coupling between $\mathbf{z}_t$ and $\mathbf{r}$ through the latent variable $\mathbf{z}_0$, despite $\mathbf{z}_t$ and $\mathbf{r}$ being conditionally independent given $\mathbf{z}_0$. This coupling is what enables semantic-conditioned generation.

The augmented probability space construction embeds the desired semantic knowledge directly into the probabilistic model. The generative task is thus transformed from learning an unconditional reverse process to learning a *conditional* reverse process, where synthesis is explicitly conditioned on a target semantic concept $\mathbf{r}$. This transformation is fundamental to understanding why our two-phase training approach is theoretically justified.

### B.2.3 SEMANTIC SUFFICIENCY AND CONDITIONAL INDEPENDENCE

The power of the augmented formulation relies on a key assumption about the semantic representation, which we formalize below.

> **Assumption 1 (Semantic Sufficiency) .** *The semantic representation $\mathbf{r} = \mathbf{f}_{rep}(\mathbf{x})$ captures sufficient information for the generative task such that, given $\mathbf{r}$, the model possesses all necessary high-level information to synthesize a corresponding sample. Formally, this means that the conditional distribution $p(\mathbf{z}_0 \mid \mathbf{r})$ concentrates on semantically-consistent latents.*

**Intuitive Understanding:** This assumption embodies the idea that our pretrained representation model $\mathbf{f}_{rep}$ (e.g., DINOv2) is *sufficiently powerful* to capture all the *high-level, conceptual* information needed for generation. To illustrate with an analogy: if $\mathbf{r}$ represents "a golden retriever running on grass," then semantic sufficiency means that knowing this $\mathbf{r}$ provides the model with all the essential semantic components—the subject (dog), category (golden retriever), action (running), and environment (grass). The model's remaining task shifts from deciding *what to generate* to focusing purely on *how to generate it*: the specific pose, fur details, lighting direction, grass texture, etc.

**Latent Space Partitioning:** More precisely, we assume there exists a partition of the latent space based on semantic content. The semantic representation $\mathbf{r}$ acts like a clustering label that groups clean latents with identical semantic meaning. We define semantic equivalence classes:

$$\mathcal{Z}_{\mathbf{r}} = \{\mathbf{z}_0 \in \mathcal{Z} : \mathbf{f}_{rep}(\mathcal{D}(\mathbf{z}_0)) = \mathbf{r}\} \tag{31}$$

For example, $\mathcal{Z}_{\text{"cat"}}$ might contain latents corresponding to "a crouching Persian cat," "a rolling orange tabby," and "a sleeping Siamese cat." Despite their vastly different visual details, all belong to the same semantic category under $\mathbf{f}_{rep}$.

**Overlap Requirement for Well-Posed Generation:** A critical consequence of semantic sufficiency is that for any two clean latents $\mathbf{z}_0, \mathbf{z}_0' \in \mathcal{Z}_{\mathbf{r}}$, their respective forward diffusion processes $p(\mathbf{z}_t \mid \mathbf{z}_0, t)$ and $p(\mathbf{z}_t \mid \mathbf{z}_0', t)$ should have *significant overlap*. This requirement ensures that conditional generation remains well-posed:

- **Without overlap:** If semantically similar $\mathbf{z}_0$ values produce completely different noisy patterns $\mathbf{z}_t$, the model becomes "confused"—it cannot learn a consistent denoising pattern for the semantic class $\mathbf{r}$.
- **With overlap:** When $\mathbf{z}_0$ values in $\mathcal{Z}_{\mathbf{r}}$ yield similar noisy distributions, the model can learn a unified denoising strategy conditioned on $\mathbf{r}$.

### B.2.4 CONDITIONAL SCORE MATCHING IN THE AUGMENTED SPACE

The central idea is to model the score of the joint conditional distribution $p(\mathbf{z}_0, \mathbf{r} \mid \mathbf{z}_t, t)$, which naturally decomposes into two meaningful components.

> **Theorem 1 (Decomposition of the Augmented Conditional Score) .** *The score of the joint conditional distribution $p(\mathbf{z}_0, \mathbf{r} \mid \mathbf{z}_t, t)$ can be decomposed into a sum of two functionally distinct scores:*
>
> $$\nabla_{\mathbf{z}_t} \log p(\mathbf{z}_0, \mathbf{r} \mid \mathbf{z}_t, t) = \underbrace{\nabla_{\mathbf{z}_t} \log p(\mathbf{z}_0 \mid \mathbf{z}_t, \mathbf{r}, t)}_{\text{Conditional Generation Score}} + \underbrace{\nabla_{\mathbf{z}_t} \log p(\mathbf{r} \mid \mathbf{z}_t, t)}_{\text{Representation Inference Score}} \tag{32}$$

*Proof.* We provide a detailed derivation of this fundamental decomposition.

**Step 1: Probabilistic factorization.** Using the chain rule of conditional probability, we can factorize the joint conditional distribution:

$$p(\mathbf{z}_0, \mathbf{r} \mid \mathbf{z}_t, t) = p(\mathbf{z}_0 \mid \mathbf{z}_t, \mathbf{r}, t) \, p(\mathbf{r} \mid \mathbf{z}_t, t) \tag{33}$$

This factorization is always valid and separates the problem into two components: generating clean latents given both noisy latents and semantic information, and inferring semantic information from noisy latents.

**Step 2: Logarithmic transformation.** Taking the natural logarithm of both sides of equation (33):

$$\log p(\mathbf{z}_0, \mathbf{r} \mid \mathbf{z}_t, t) = \log \left[ p(\mathbf{z}_0 \mid \mathbf{z}_t, \mathbf{r}, t) \, p(\mathbf{r} \mid \mathbf{z}_t, t) \right] \tag{34}$$

$$= \log p(\mathbf{z}_0 \mid \mathbf{z}_t, \mathbf{r}, t) + \log p(\mathbf{r} \mid \mathbf{z}_t, t) \tag{35}$$

where we used the logarithm property $\log(ab) = \log a + \log b$.

**Step 3: Gradient computation.** Applying the gradient operator $\nabla_{\mathbf{z}_t}$ with respect to the noisy latent $\mathbf{z}_t$ to both sides of equation (35):

$$\nabla_{\mathbf{z}_t} \log p(\mathbf{z}_0, \mathbf{r} \mid \mathbf{z}_t, t) = \nabla_{\mathbf{z}_t} \left[ \log p(\mathbf{z}_0 \mid \mathbf{z}_t, \mathbf{r}, t) + \log p(\mathbf{r} \mid \mathbf{z}_t, t) \right] \tag{36}$$

$$= \nabla_{\mathbf{z}_t} \log p(\mathbf{z}_0 \mid \mathbf{z}_t, \mathbf{r}, t) + \nabla_{\mathbf{z}_t} \log p(\mathbf{r} \mid \mathbf{z}_t, t) \tag{37}$$

where we used the linearity of the gradient operator: $\nabla(f + g) = \nabla f + \nabla g$.

**Step 4: Functional interpretation.** The resulting decomposition has clear functional meaning:

- $\nabla_{\mathbf{z}_t} \log p(\mathbf{z}_0 \mid \mathbf{z}_t, \mathbf{r}, t)$ represents the *Conditional Generation Score*: given both noisy input $\mathbf{z}_t$ and semantic target $\mathbf{r}$, how should we move in latent space to increase the likelihood of the clean latent $\mathbf{z}_0$?

- $\nabla_{\mathbf{z}_t} \log p(\mathbf{r} \mid \mathbf{z}_t, t)$ represents the *Representation Inference Score*: given only noisy input $\mathbf{z}_t$, how should we move in latent space to increase the likelihood of the semantic representation $\mathbf{r}$?

This completes the proof of the score decomposition in equation (32). $\square$

---

**Corollary 1 (Functional Interpretation of Score Components) .** *Thm. 1 provides the central theoretical insight of our work. The total learning objective is a linear superposition of two functionally distinct tasks:*

1. ***Conditional Generation Score**: The term $\nabla_{\mathbf{z}_t} \log p(\mathbf{z}_0 \mid \mathbf{z}_t, \mathbf{r}, t)$ corresponds to the **R2G** (**Representation-to-Generation**) circuit. It addresses the pure synthesis problem: given a noisy latent $\mathbf{z}_t$ and the ground-truth semantic concept $\mathbf{r}$, compute the score vector towards the clean latent $\mathbf{z}_0$.*

2. ***Representation Inference Score**: The term $\nabla_{\mathbf{z}_t} \log p(\mathbf{r} \mid \mathbf{z}_t, t)$ corresponds to the **L2R** (**Latent-to-Representation**) circuit. It addresses the semantic inference problem: given only a noisy latent $\mathbf{z}_t$, compute the score vector that increases the likelihood of the underlying semantic representation being $\mathbf{r}$.*

---

### B.2.5 EMERGENCE OF THE TWO-PHASE TRAINING FRAMEWORK FROM THE THEORY

A naive attempt to train a single, monolithic network $\mathbf{F}_\theta$ to approximate the joint score in (32) would re-entangle the two objectives, leading to optimization conflicts. A more principled approach, suggested by the decomposition itself, is a curriculum learning strategy that addresses the two scores in a structured sequence. This naturally gives rise to the **ERW** framework.

---

**Lemma 1 (Phase 1: Representation Warmup as Boundary Condition Matching) .** *The first phase of* ERW*, the representation warmup, can be interpreted as learning the **Representation Inference Score** at the clean boundary condition, i.e., at $t = 0$.*

---

*Proof.* We provide a detailed derivation showing how the warmup phase corresponds to boundary condition matching.

**Step 1: Analysis at the boundary condition.** At $t = 0$, the forward diffusion process gives us $\mathbf{z}_t = \mathbf{z}_0$ (no noise added). The Representation Inference Score becomes:

$$\nabla_{\mathbf{z}_t} \log p(\mathbf{r} \mid \mathbf{z}_t, t) \big|_{t=0} = \nabla_{\mathbf{z}_0} \log p(\mathbf{r} \mid \mathbf{z}_0) \tag{38}$$

**Step 2: Simplification using the semantic constraint.** From equation (19), we have:

$$p(\mathbf{r} \mid \mathbf{z}_0) = \delta \left( \mathbf{r} - \mathbf{f}_{\text{rep}} \left( \mathcal{D}_{\boldsymbol{\theta}_{\text{VAE}}}(\mathbf{z}_0) \right) \right) \tag{39}$$

This is a Dirac delta function, which means the score is not well-defined in the classical sense. However, we can interpret this in terms of the desired functional behavior.

**Step 3: Functional interpretation and approximation.** In practice, we approximate the deterministic relationship through a learned mapping. The warmup objective is:

$$\mathcal{L}_{\text{warmup}} = \mathbb{E}_{\mathbf{x} \sim p_{\text{data}}} \Big[ \ell_{\text{NT-Xent}} \big( \mathcal{T}_{\boldsymbol{\theta}}(\mathcal{R}_{\boldsymbol{\theta}_{\text{L2R}}}(\mathcal{H}_{\boldsymbol{\theta}_{\text{VAE}}}(\mathbf{x}))), \ \mathbf{f}_{\text{rep}}(\mathbf{x}) \big) \Big] \tag{40}$$

where $\mathcal{R}_{\boldsymbol{\theta}_{\text{L2R}}}$ is the L2R circuit that we train to approximate the mapping $\mathbf{z}_0 \mapsto \mathbf{r}$.

**Step 4: Connection to boundary condition.** Optimizing NT-Xent at $t=0$ serves as boundary-condition matching for representation alignment. Specifically, we want:

$$\mathcal{R}_{\boldsymbol{\theta}_{\text{L2R}}}(\mathbf{z}_0) \approx \mathbf{f}_{\text{rep}}(\mathcal{D}_{\boldsymbol{\theta}_{\text{VAE}}}(\mathbf{z}_0)) = \mathbf{r} \tag{41}$$

This provides a strong "semantic anchor" for the model at $t = 0$, ensuring that the L2R circuit learns to extract meaningful semantic representations from clean latents under the same contrastive objective used in Phase 2.

**Step 5: Extension to $t > 0$.** Once the boundary condition is satisfied, the L2R circuit can be expected to generalize to noisy inputs $\mathbf{z}_t$ for $t > 0$, providing a foundation for the representation inference score at all time steps. $\qquad\square$

---

> **Lemma 2 (Phase 2: Guided Synthesis as Joint Score Optimization) .** *The second phase of* ERW, *guided synthesis, corresponds to learning the full joint score, where the two components from* Thm. 1 *are learned concurrently under a curriculum.*

---

*Proof.* We demonstrate how Phase 2 implements joint score optimization through a carefully designed curriculum.

**Step 1: Phase 2 objective decomposition.** After the warmup phase, the L2R circuit $\mathcal{R}_{\boldsymbol{\theta}_{\text{L2R}}}$ is a competent representation extractor. The Phase 2 total loss is:

$$\mathcal{L}_{\text{total}} = \mathcal{L}_{\text{diffusion}} + \lambda_{\text{train}}(k) \cdot \mathcal{L}_{\text{align}} \tag{42}$$

where:

$$\mathcal{L}_{\text{diffusion}} = \mathbb{E}_{t, \mathbf{z}_t, \mathbf{z}_0} \Big[ w(t) \left\| \mathbf{F}_{\boldsymbol{\theta}}(\mathbf{z}_t, t) - \nabla_{\mathbf{z}_t} \log p(\mathbf{z}_0 \mid \mathbf{z}_t, t) \right\|^2 \Big] \tag{43}$$

$$\mathcal{L}_{\text{align}} = \mathbb{E}_{\mathbf{z}_t, \mathbf{r}} \Big[ \ell_{\text{align}} \big( \mathcal{R}_{\boldsymbol{\theta}_{\text{L2R}}}(\mathbf{z}_t, t), \ \mathbf{r} \big) \Big] \tag{44}$$

**Step 2: Connection to the score decomposition.** From Thm. 1, the joint conditional score decomposes as:

$$\nabla_{\mathbf{z}_t} \log p(\mathbf{z}_0, \mathbf{r} \mid \mathbf{z}_t, t) = \nabla_{\mathbf{z}_t} \log p(\mathbf{z}_0 \mid \mathbf{z}_t, \mathbf{r}, t) + \nabla_{\mathbf{z}_t} \log p(\mathbf{r} \mid \mathbf{z}_t, t) \tag{45}$$

Our Phase 2 objective should be interpreted as shaping these two functional components via practical surrogate losses: the standard diffusion loss $\mathcal{L}_{\text{diffusion}}$ for generation and the alignment loss $\mathcal{L}_{\text{align}}$ for representation, rather than claiming exact equality to the joint score at all times. In practice, we instantiate $\ell_{\text{align}}$ as a contrastive objective (e.g., NT-Xent) with in-batch negatives.

**Step 3: Curriculum learning analysis.** The training-schedule-dependent weighting $\lambda_{\text{train}}(k)$ creates a curriculum that balances the two objectives:

- **Early in Phase 2** (large $\lambda_{\text{train}}(k)$):

$$\mathcal{L}_{\text{total}} \approx \lambda_{\text{train}}(k) \cdot \mathcal{L}_{\text{align}} + \mathcal{L}_{\text{diffusion}} \tag{46}$$

  The optimization is strongly guided to maintain semantic consistency on noisy inputs, reinforcing the L2R circuit's ability to extract representations from $\mathbf{z}_t$ for $t > 0$.

- **Late in Phase 2** (small $\lambda_{\text{train}}(k)$):

$$\mathcal{L}_{\text{total}} \approx \mathcal{L}_{\text{diffusion}} \tag{47}$$

  The L2R circuit is assumed to be robust, and optimization focus shifts to perfecting the full score matching. This allows the R2G circuit to learn the Conditional Generation Score while relying on stable, high-quality representations from the L2R circuit.

$\qquad\square$

## C  ANALYSIS DETAILS

### C.1  CKNNA METRIC DETAILS

**CKNNA** (Centered Kernel Nearest-Neighbor Alignment) is a *relaxed version* of the popular Centered Kernel Alignment (CKA; Kornblith et al. 2019) that mitigates the strict definition of alignment. We generally follow the notations in the original paper for an explanation (Huh et al., 2024).

First, CKA have measured *global* similarities of the models by considering all possible data pairs:

$$\mathrm{CKA}(\mathbf{K}, \mathbf{L}) = \frac{\mathrm{HSIC}(\mathbf{K}, \mathbf{L})}{\sqrt{\mathrm{HSIC}(\mathbf{K}, \mathbf{K})\mathrm{HSIC}(\mathbf{L}, \mathbf{L})}}, \tag{48}$$

where $\mathbf{K}$ and $\mathbf{L}$ are two kernel matrices computed from the dataset using two different networks. Specifically, it is defined as $\mathbf{K}_{ij} = \kappa(\phi_i, \phi_j)$ and $\mathbf{L}_{ij} = \kappa(\psi_i, \psi_j)$ where $\phi_i, \phi_j$ and $\psi_i, \psi_j$ are representations computed from each network at the corresponding data $\mathbf{x}_i, \mathbf{x}_j$ (respectively). By letting $\kappa$ as a inner product kernel, HSIC is defined as

$$\mathrm{HSIC}(\mathbf{K}, \mathbf{L}) = \frac{1}{(n-1)^2} \Big( \sum_i \sum_j \big( \langle \phi_i, \phi_j \rangle - \mathbb{E}_l[\langle \phi_i, \phi_l \rangle] \big) \big( \langle \psi_i, \psi_j \rangle - \mathbb{E}_l[\langle \psi_i, \psi_l \rangle] \big) \Big). \tag{49}$$

CKNNA considers a relaxed version of Eq. (48) by replacing $\mathrm{HSIC}(\mathbf{K}, \mathbf{L})$ into $\mathrm{Align}(\mathbf{K}, \mathbf{L})$, where $\mathrm{Align}(\mathbf{K}, \mathbf{L})$ computes Eq. (49) only using a $k$-nearest neighborhood embedding in the datasets:

$$\mathrm{Align}(\mathbf{K}, \mathbf{L}) = \frac{1}{(n-1)^2} \Big( \sum_i \sum_j \alpha(i, j) \big( \langle \phi_i, \phi_j \rangle - \mathbb{E}_l[\langle \phi_i, \phi_l \rangle] \big) \big( \langle \psi_i, \psi_j \rangle - \mathbb{E}_l[\langle \psi_i, \psi_l \rangle] \big) \Big), \tag{50}$$

where $\alpha(i, j)$ is defined as

$$\alpha(i, j; k) = \mathbb{1}[i \neq j \text{ and } \phi_j \in \mathrm{knn}(\phi_i; k) \text{ and } \psi_j \in \mathrm{knn}(\psi_i; k)], \tag{51}$$

so this term only considers $k$-nearest neighbors at each $i$. In this paper, we randomly sample 10,000 images in the validation set in ImageNet (Deng et al., 2009) and report CKNNA with $k = 10$ based on observation in Huh et al. (2024) that smaller $k$ shows better a better alignment.

### C.2  DESCRIPTION OF PRETRAINED VISUAL ENCODERS

- **MoCov3** (Chen et al., 2021) studies empirical study to train MoCo (He et al., 2020; Chen et al., 2020b) on vision transformer and how they can be scaled up.
- **CLIP** (Radford et al., 2021) proposes a contrastive learning scheme on large image-text pairs.
- **DINOv2** (Oquab et al., 2024) proposes a self-supervised learning method that combines pixel-level and patch-level discriminative objectives by leveraging advanced self-supervised techniques and a large pre-training dataset.

## D  HYPERPARAMETER AND MORE IMPLEMENTATION DETAILS

### D.1  HYPERPARAMETER TUNING

We adopt a bisection-style search to determine the key hyperparameters for ERW, specifically the ERW *Depth* (i.e., which early layers to initialize), the *Projection Depth*, and the initial value of $\lambda$ in Eq. (14). To keep the search computationally manageable, we do the following for each candidate hyperparameter setting:

(a) We run a short warmup stage for 10k iterations, followed by 20k iterations of main diffusion training.

(b) To evaluate performance quickly, we reduce the sampling steps from the usual 250 to 50 and generate only 10k samples (instead of 50k) to compute a preliminary FID score.

This procedure substantially reduces the search cost while retaining sufficient fidelity to guide hyperparameter choices. In practice, around three to five such tests suffice to converge upon near-optimal settings for ERW Depth, Projection Depth, and $\lambda$, enabling both efficient training and high-quality generation.

**Further implementation details.** We implement our model based on the original SiT implementation (Ma et al., 2024). Throughout the experiments, we use the exact same structure as DiT (Peebles & Xie, 2023) and SiT (Ma et al., 2024). We use AdamW (Kingma, 2015; Loshchilov, 2017)

Table 6: Hyperparameter setup.

|  | Figure 1,2,3 | Table 3,4 (SiT-B) | Table 1,2,5 (SiT-XL) |
|---|---|---|---|
| **Architecture** | | | |
| Input dim. | $32{\times}32{\times}4$ | $32{\times}32{\times}4$ | $32{\times}32{\times}4$ |
| Num. layers | 28 | 12 | 24 |
| Hidden dim. | 1,152 | 768 | 1,152 |
| Num. heads | 16 | 12 | 16 |
| **ERW** | | | |
| $\text{sim}(\cdot,\cdot)$ | NT-Xent | NT-Xent | NT-Xent |
| Encoder $f(\mathbf{x})$ | DINOv2-B | DINOv2-B | DINOv2-B |
| **Optimization** | | | |
| Batch size | 256 | 256 | 256 |
| Optimizer | AdamW | AdamW | AdamW |
| lr | 0.0001 | 0.0001 | 0.0001 |
| $(\beta_1, \beta_2)$ | (0.9, 0.999) | (0.9, 0.999) | (0.9, 0.999) |
| **Interpolants** | | | |
| $\alpha_t$ | $1-t$ | $1-t$ | $1-t$ |
| $\sigma_t$ | $t$ | $t$ | $t$ |
| $w_t$ | $\sigma_t$ | $\sigma_t$ | $\sigma_t$ |
| Training objective | v-prediction | v-prediction | v-prediction |
| Sampler | Euler-Maruyama | Euler-Maruyama | Euler-Maruyama |
| Sampling steps | 250 | 250 | 250 |
| Guidance | - | - | - |

Table 7: **Impact of Training Tricks in ERW**. Using the SD-VAE Rombach et al. (2022), ERW achieves an FID of 55.6 at 50K training steps on ImageNet class-conditional generation. This table illustrates how each training trick incrementally improves the FID, demonstrating that advanced design techniques enhance the original DiT performance.

| Training Trick | Training Step | FID-50k↓ |
|---|---|---|
| *Representation Alignment Loss* | | |
| + REPA (Yu et al., 2024) | 50K | 78.2 |
| *Architecture Improvements* | | |
| + Rotary Pos Embed (Su et al., 2024) | 50K | 73.6 |
| *Initialization* | | |
| + ERW (Ours) | 50K | 51.7 |

with constant learning rate of 1e-4, $(\beta_1, \beta_2) = (0.9, 0.999)$ without weight decay. To speed up training, we use mixed-precision (fp16) with gradient clipping at norm 1.0. We also pre-compute compressed latent vectors from raw pixels via stable diffusion VAE (Rombach et al., 2022) and use these latent vectors. Because of this, we do not apply any data augmentation, but we find this does not lead to a big difference, as similarly observed in EDM2 (Karras et al., 2024). We also use `stabilityai/sd-vae-ft-ema` decoder for decoding latent vectors to images. For MLP used for a projection, we use three-layer MLP with SiLU activations (Elfwing et al., 2018). We provide a detailed hyperparameter setup in Table 6.

**Pretrained encoders.** For MoCov3-B and -L models, we use the checkpoint in the implementation of RCG (Li et al., 2024);[1] for other checkpoints, we use their official checkpoints released in their official implementations. To adjust a different number of patches between the diffusion transformer and the pretrained encoder, we interpolate positional embeddings of pretrained encoders.

**Sampler.** For sampling, we use the Euler-Maruyama sampler with the SDE with a diffusion coefficient $w_t = \sigma_t$. We use the last step of the SDE sampler as 0.04, and it gives a significant improvement, similar to the original SiT paper (Ma et al., 2024).

**Training Tricks.** We explore the influence of various training techniques on ERW's performance. Notably, we observe performance improvements when incorporating Rotary Positional Embeddings (Su et al., 2024).

---

[1] https://github.com/LTH14/rcg

## E EVALUATION DETAILS

We strictly follow the setup and use the same reference batches of ADM (Dhariwal & Nichol, 2021) for evaluation, following their official implementation.[2] We use 8×NVIDIA H800 80GB GPUs for evaluation and enable tf32 precision for faster generation, and we find the performance difference is negligible to the original fp32 precision.

In what follows, we explain the main concept of metrics that we used for the evaluation.

- **FID** (Heusel et al., 2017) measures the feature distance between the distributions of real and generated images. It uses the Inception-v3 network (Szegedy et al., 2016) and computes distance based on an assumption that both feature distributions are multivariate gaussian distributions.

- **sFID** (Nash et al., 2021) proposes to compute FID with intermediate spatial features of the Inception-v3 network to capture the generated images' spatial distribution.

- **IS** (Salimans et al., 2016) also uses the Inception-v3 network but use logit for evaluation of the metric. Specifically, it measures a KL-divergence between the original label distribution and the distribution of logits after the softmax normalization.

- **Precision and recall** (Kynkäänniemi et al., 2019) are based on their classic definitions: the fraction of realistic images and the fraction of training data manifold covered by generated data.

## F BASELINES

In what follows, we explain the main idea of baseline methods that we used for the evaluation.

- **ADM** (Dhariwal & Nichol, 2021) improves U-Net-based architectures for diffusion models and proposes classifier-guided sampling to balance the quality and diversity tradeoff.

- **VDM++** (Kingma & Gao, 2024) proposes a simple adaptive noise schedule for diffusion models to improve training efficiency.

- **Simple diffusion** (Hoogeboom et al., 2023) proposes a diffusion model for high-resolution image generation by exploring various techniques to simplify a noise schedule and architectures.

- **CDM** (Ho et al., 2022) introduces cascaded diffusion models: similar to progressiveGAN (Karras et al., 2018), it trains multiple diffusion models starting from the lowest resolution and applying one or more super-resolution diffusion models for generating high-fidelity images.

- **LDM** (Rombach et al., 2022) proposes latent diffusion models by modeling image distribution in a compressed latent space to improve the training efficiency without sacrificing the generation performance.

- **U-ViT** (Bao et al., 2023) proposes a ViT-based latent diffusion model that incorporates U-Net-like long skip connections.

- **DiffiT** (Hatamizadeh et al., 2024) proposes a time-dependent multi-head self-attention mechanism for enhancing the efficiency of transformer-based image diffusion models.

- **MDTv2** (Gao et al., 2023) proposes an asymmetric encoder-decoder scheme for efficient training of a diffusion-based transformer. They also apply U-Net-like long-shortcuts in the encoder and dense input-shortcuts in the decoder.

- **MaskDiT** (Zheng et al., 2024) proposes an asymmetric encoder-decoder scheme for efficient training of diffusion transformers, where they train the model with an auxiliary mask reconstruction task similar to MAE (He et al., 2022).

- **SD-DiT** (Zhu et al., 2024) extends MaskdiT architecture but incorporates self-supervised discrimination objective using a momentum encoder.

- **DiT** (Peebles & Xie, 2023) proposes a pure transformer backbone for training diffusion models based on proposing AdaIN-zero modules.

- **SiT** (Ma et al., 2024) extensively analyzes how DiT training can be efficient by moving from discrete diffusion to continuous flow-based modeling.

- **REPA** (Yu et al., 2024) proposes a representation alignment method for diffusion models by aligning the representation of the diffusion model with a pretrained encoder.

---

[2]https://github.com/openai/guided-diffusion/tree/main/evaluations

