# OpenReview forum: "Efficient Generative Model Training via Embedded Representation Warmup"
_ICLR.cc/2026/Conference — ICLR 2026 Conference Withdrawn Submission_

### Official Review · Reviewer_mjRP · 2025-10-24

**Soundness:** 2
**Presentation:** 2
**Contribution:** 1
**Rating:** 2
**Confidence:** 4

**Summary:**

This paper identifies that the learning for high-level semantic understanding ("what" to generate) and low-level synthesis details ("how" to generate) is entangled in diffusion model training and argues that forcing diffusion models to learn both simultaneously from scratch in an end-to-end fashion is suboptimal. To solve this, a two-phase training framework is proposed to disentangle those two components, where the first phase aligns the semantic representation part of the network with a pretrained encoder, and the second phase fits the whole network with a standard diffusion loss and the alignment loss. The proposed method is evaluated on ImageNet256 image generation task, achieving a 11.5x training speedup while maintaining comparable performance to previous method.

**Strengths:**

The observation of the entanglement between semantic representation learning and generative refinement in diffusion model training is interesting, which serves as a natural motivation for the proposed method.

**Weaknesses:**

1. The definition of the joint conditional score in Eq (6) is incorrect. In a score matching objective, the target score to be matched should be the score of a probability distribution over $z_t$ rather than $z_0$. Hence, I believe Eq (6) should be $\nabla\_{z_t} \log p(z_t, r|z_0, t)=\nabla\_{z_t} \log p(z_t|z_0, r, t)+\nabla\_{z_t} \log p(r|z_0, t)$.
2. The generation process defined in Eq (10) is incorrect. According to Eq (11), $\mathcal{G}\_{\theta\_{R2G}}(r_t, t)$ approximates the conditional generation score. However, in the denoising process of a diffusion model, this score does not directly correspond to a denoised latent $z_{t-\Delta  t}$. The correct way to obtain $z_{t-\Delta  t}$ is to solve the reverse SDE of the diffusion process using $\mathcal{G}\_{\theta\_{R2G}}(r_t, t)$.
3. The novelty of the proposed method is limited and does not meet the acceptance bar in my opinion. The core techniques in the proposed two-phase training framework (alignment and generation) are directly taken from existing methods in the literature.
4. Given the incremental nature of the proposed method, it is not sufficient to demonstrate the advantages of the method on the ImageNet image generation task alone. I would expect to see a more compressive empirical evaluation on more complicated tasks (e.g., text-to-image generation, video generation).
5. The proposed method requires a pretrained encoder for alignment, which may not always be available (e.g., in video generation tasks).
6. Limitations of the proposed method are not discussed in the paper. Also, Section 5 is called "Conclusion and Future Work", but future work is not discussed there.

**Questions:**

1. What are $\mathcal{T}\_\theta$ and $l\_{NT-Xent}$ in the alignment loss in Eq (13)? These terms are not introduced in the text in Section 3.3.
2. Given the incremental nature of the proposed method, could the authors demonstrate the advantages of the proposed method on more complicated tasks, such as text-to-image generation and video generation?
3. Please discuss the limitations of the proposed method in the paper.

---

### Official Review · Reviewer_H4po · 2025-10-27

**Soundness:** 3
**Presentation:** 2
**Contribution:** 3
**Rating:** 6
**Confidence:** 2

**Summary:**

A method for improving the training and performance of diffusion models is proposed. The idea of the method is to align a diffusion model's denoising process with the "representation first, generation later" scenario, by adding a loss to match the early stage of the denoising process with the representations obtained by pretrained image encoders. The proposed method achieves better generation performance with a smaller number of training epochs.

**Strengths:**

- The method is simple, and its background sounds technically reasonable.
- The experiment well supports the utility of the method.

**Weaknesses:**

Although the paper is basically not difficult to follow, the explanations of some important concepts are skipped. It is probably due to the length limit, and appropriate references are presented, but for the completeness of the paper they should be more integrated into the reasoning, so that readers who are not necessarily familiar with all the references can follow. Most importantly, the core of the proposed method, Eq. (13), fails to define $\ell_\text{NT-Xent}$ and $T_\theta$. I guess these are defined somewhere in the appendix, but the definitions should be in the main text (sorry if I am missing something). Also, $\mathbf{f}_\text{rep}$ is defined as a map $\mathcal{X} \to \mathcal{R}$, but in my understanding $\mathcal{R}$ is used as another map and not as some space.

**Questions:**

Line 323:

> This makes the training-time decomposition mirror the sampling-time loop: first representation (L2R), then generation (R2G).

Do you have any direct evidence of this claim? I guess some of the results in Section 4.4 are relevant but could not follow them well. More detailed, clear explanation of the implication of these results will be highly appreciated.

---

### Official Review · Reviewer_k4cc · 2025-10-31

**Soundness:** 4
**Presentation:** 4
**Contribution:** 3
**Rating:** 6
**Confidence:** 4

**Summary:**

This paper proposes Embedded Representation Warmup (ERW), a two-phase diffusion model training strategy aimed at decoupling semantic understanding from visual synthesis. Latent-to-Representation (L2R) aligns early layers with a pre-trained visual encoder to build a strong semantic foundation and Representation-to-Generation (R2G) focus on visual synthesis with a gradually decreasing alignment weight. Experiments show that decoupling strategy leads to faster convergence (11.5x) and higher generation quality compared to naively using alignment loss throughout the training.

**Strengths:**

- The paper shows strong experimental results, with clear improvements in convergence speed using the warmup strategy (with its relatively small overhead). Also, the ablation studies are detailed and convincing.
- The paper is clearly written and easy to follow, and the figures and tables are well-organized with detailed captions that make the results easy to understand.
- The inclusion of a theoretical framework adds meaning beyond empirical demonstrations.

**Weaknesses:**

- It would be helpful to include an ablation study on the warmup phase. For example, how sensitive is the method to the warmup duration? Is a short warmup sufficient for faster convergence, or are there empirical signals indicating when to begin the R2G phase?

- Can the proposed method be easily extended to text-to-image generation tasks?

- Experiments on higher-resolution (e.g., 512×512) and on DiTs would further strengthen the experimental validation.

**Questions:**

- If we also align noisy inputs (instead of only clean images) during the warmup phase, would it lead to better semantic representation learning? Or would it slow down the lightweight warmup stage?

- The paper shows consistent performance across different DINOv2 variants, even without the largest encoder. Since previous work (e.g., REPA) found that stronger encoders lead to better generation quality, is there any analysis on why this method works well with smaller ones, and whether it can be effectively applied to domains without strong pre-trained encoders (e.g., video)?

---

### Official Review · Reviewer_7LHf · 2025-10-31

**Soundness:** 2
**Presentation:** 3
**Contribution:** 3
**Rating:** 4
**Confidence:** 3

**Summary:**

Representation Learning is a core element of machine learning research. The paper proposes a a decoupled pipeline for more efficient generative modeling. Decoupling the generative process has many potential advantages and the paper claims that the two step process has many advantages. This is supported by faster training and very positive comparisons even to very recent baselines from ICLR 2025 (REPA).

The paper provides extensive experiments, is easy to read and addresses a core problem in machine learning research.

**Strengths:**

The paper is nicely written, addresses an important problem, has many recent baselines as comparisons. While the paper claims SOTA results the contribution would probably be less about being SOTA but by shifting the field towards a decoupled approach and that decoupling is the right strategy for representation learning.
This second part is still not absolutely clear to me e.g. it might depend significantly on the framework which is used i.e. viewing representation learning as pixel to latent, Latent-to-Representation and Representation-to-Generation.

**Weaknesses:**

The paper provides extensive experiments yet it is of course limited what one can do and in how many settings one can investigate the decoupling.

In particular I would be interested to see if the framework which is used i.e. viewing representation learning as pixel to latent, Latent-to-Representation and Representation-to-Generation depends significantly on the dimension of the latent and still performing representation learning with some form of information bottleneck and low dimensional representations as well as the reconstruction based training.

**Questions:**

In particular I would be interested to see if the framework which is used i.e. viewing representation learning as pixel to latent, Latent-to-Representation and Representation-to-Generation depends significantly on the dimension of the latent and still performing representation learning with some form of information bottleneck and low dimensional representations as well as the reconstruction based training.

---

### Note · Authors · 2025-11-13

I have read and agree with the venue's withdrawal policy on behalf of myself and my co-authors.